# Identification of Key Carbon Emission Industries and Emission Reduction Control Based on Complex Network of Embodied Carbon Emission Transfers: The Case of Hei-Ji-Liao, China

**DOI:** 10.3390/ijerph20032603

**Published:** 2023-01-31

**Authors:** Shaonan Shan, Yulong Li, Zicheng Zhang, Wei Zhu, Tingting Zhang

**Affiliations:** 1School of Business, Shenyang University, Shenyang 110064, China; 2School of Information Management, Nanjing University, Nanjing 210023, China; 3Institute of Industrial and Economic Policy, Beijing Economic and Technological Development Zone (BDA), Beijing 100070, China; 4School of Public Finance and Taxation, Capital University of Economics and Business, Beijing 100070, China

**Keywords:** embodied carbon emissions, industrial sectors, input–output model, carbon reduction control, social network analysis

## Abstract

Similar to the problems surrounding carbon transfers that exist in international trade, there are severe carbon emission headaches in regional industrial systems within countries. It is essential for emission reduction control and regional industrial restructuring to clarify the relationship of carbon emissions flows between industrial sectors and identify key carbon-emitting industrial sectors. Supported by the input–output model (I-O model) and social network analysis (SNA), this research adopts input–output tables (2017), energy balance sheets (2021) and the energy statistics yearbooks (2021) of the three Chinese provinces of Hei-Ji-Liao to construct an Embodied carbon emission transfer network (ECETN) and determine key carbon-emitting industrial sectors with a series of complex network measurement indicators and analysis methods. The key abatement control pathways are obtained based on the flow relationships between the chains in the industrial system. The results demonstrate that the ECETNs in all three provinces of Hei-Ji-Liao are small-world in nature with scale-free characteristics (varying according to the power function). The key carbon emission industry sectors in the three provinces are identified through centrality, influence, aggregation and diffusion, comprising coal mining, the chemical industry, metal products industry, machinery manufacturing and transportation in Liaoning Province; coal mining, non-metal mining, non-metal products, metal processing and the electricity industry in Jilin Province; and agriculture, metal processing and machinery manufacturing in Heilongjiang. Additionally, key emission reduction control pathways in the three provinces are also identified based on embodied carbon emission flow relationships between industry sectors. Following the above findings, corresponding policy recommendations are proposed to tackle the responsibility of carbon reduction among industrial sectors in the province. Moreover, these findings provide some theoretical support and policy considerations for policymakers.

## 1. Introduction

Global warming, which has become a common challenge for the whole world, is not only causing ecological damage but also threatening the survival and development of mankind. From the United Nations Framework Convention on Climate Change (UNFCCC) in 1992 to the Paris Agreement in 2015, the international community has reached a high level of consensus on reducing CO_2_ and other greenhouse gas emissions and implementing green and low-carbon development [1]. China’s economic growth has attracted worldwide attention since its reform and opening up, while it has consumed a large amount of energy. In the face of this situation, China proposed the “2030 China Emission Reduction Target” in 2015 to reduce CO_2_ emissions per unit of GDP by 60% to 65% in 2030 compared to 2005 and reach the peak of CO_2_ emissions as early as possible. In 2020, China went a step further and claimed that carbon emissions will peak by 2030 and the economy will become carbon neutral by 2060. China has been working towards this goal [2].

However, it is not easy for China to reduce emissions [3]. From one perspective, over 90% of China’s total carbon emissions come from industrial production and the Chinese government has introduced a series of policies to control the reduction of direct carbon emissions from industries where energy is the main driver, so as to achieve its emissions reduction targets as soon as possible [4]. However, it is challenging to optimize and upgrade the industrial structure to green energy and low-carbon in a short period of time and all industrial sectors in the industrial system are interdependent and integrated [5]. There are complex production and consumption activities between industrial sectors, leading to large amounts of direct carbon emissions that are not only used to supply their own demand for final products [6], but also flow into other industrial sectors in the form of Embodied carbon emissions along the industrial chain and the flow of intermediate products. In this way, a complex network of carbon emission flows is formed. Therefore, a single abatement policy is not enough, as illustrated in Figure 1. From another perspective, China is a vast country and the diversity of its geographical location and resource endowments has resulted in a wide range of different levels of development and importance of industrial sectors in different regions [7]. It is only through a diversified approach that low carbon emission reduction policies can be better suited to the industrial development of different regions [8].

Therefore, the relationship between Embodied carbon emission flows in different regions and different industrial sectors should be researched to realize China’s economic transformation toward green and low-carbon development [9]. Identifying the key carbon-emitting industrial sectors in the Embodied carbon flow network and analyzing the key Embodied carbon emission pathways will assist policymakers in formulating industrial restructuring plans [10], clarifying the implementation steps of emission reduction targets, improving energy utilization [3,11], promoting cleaner production in enterprises and mitigating overall environmental pollution emissions [12].

In international research, the concept of “Embodied carbon emissions” came to prominence in the 1970s. The first to introduce the concept was the International Federation of Institutes of Advanced Studies (IFIAS) Working Group on Energy Analysis. This concept was employed to measure the total amount of a resource consumed directly and indirectly throughout the industrial process of a product or service [13,14]. In principle, any resource name can be attached after “Embodied” (e.g., Embodied carbon and Embodied energy) [15,16,17]. As Embodied accounting is relatively similar to input–output economics, many of the concepts in input–output models can be applied to the Embodied analysis. Hence, the concept of “Embodied carbon emissions” is the sum of the direct and indirect carbon emissions emitted by goods and services throughout the production process [18]. The nature of economic activity suggests that Embodied carbon emissions are a crucial element of CO_2_ emissions in the economic system [19]. Therefore, it is necessary to analyze the structural characteristics of the Embodied carbon flow network and identify the key carbon-emitting industrial sectors in the industrial system and the key Embodied carbon flow pathways [20]. This will help design scientifically sound inter-industry synergistic emissions reduction policies and lay the foundation for better and cleaner production in industrial sectors and improved responsibility towards carbon reduction.

The current research on Embodied carbon emissions has been conducted from two main perspectives: the trade level and the industry level. In the case of international trade, trade liberalization has a vital impact on greenhouse gas emissions. International trade has experienced unprecedented growth since the Second World War, with global trade increasing 33-fold between 1950 and 2012 and the share of global trade in GDP increasing from 5.5% to 21%. Simultaneously, greenhouse gas (GHG) emissions have risen significantly. This issue is increasingly being explored in depth by scholars. Antweiller et al. [21] used a general equilibrium model of trade and environment to test the effect of international trade opening on SO_2_ concentrations. Meanwhile, Grether et al. [22] performed a qualitative analysis to demonstrate the impact of international trade on global SO_2_ emissions. These studies reveal that trade openness has led to some environmental improvements in some areas. Nonetheless, they ignore the impact of trade on greenhouse gas emissions. Based on this study, Cole and Elliott [23] analyzed the impact of trade on four environmental indicators, such as CO_2_. Using data on CO_2_ from 32 developed and developing countries in 1975–1995, they concluded that trade openness leads to an increase in CO_2_ emissions and the scale effect is much larger than the technology effect. Frankel and Rose [24] examined open relationships for seven environmental quality indicators including CO_2_ for a given level of per capita income in several countries, with CO_2_ data from 150 countries. The results implied that trade growth has a negative impact on CO_2_ emissions. Since CO_2_ emissions are global and some of the costs will be borne by foreigners, there will be no strict CO_2_ emission limits in their own countries. In other words, most studies support the conclusion that international trade increases CO_2_ emissions.

Regarding domestic inter-regional trade, numerous studies have revealed that, as international economic development becomes uneven, less developed regions and high-carbon polluting industries from developed regions are transferred. In the case of China, there is a spatial shift in carbon emissions from the east to the central and western regions. Wang et al. [25] measured carbon emissions and inter-provincial carbon transfers in multiple provinces in China with a multi-regional input–output model. The results unveil that the provinces in China with the largest net carbon transfer out are located in the eastern coastal, southern coastal and Beijing-Tianjin-Hebei regions and the provinces with the largest net carbon transfer in are located in the northwest region. Wang et al. [14] argued that differences in regional net carbon transfers are related to regional economic development strategies, regional division of labor and regional industrial structure. Chen et al. [26] investigated the issue of inter-regional carbon equity and concluded that there is carbon inequity in regional trade, with Embodied net carbon transfer in regions mostly located in less developed regions and Embodied carbon transfer out regions mostly located in economically developed regions. The above studies demonstrate the patterns of Embodied carbon emissions transfer between different regions and identify the main contributors to Embodied carbon emissions, contributing to the formulation of regional carbon reduction policies.

In addition to the trade aspect, there are more complex relationships between industries in terms of carbon emissions flows. Wang et al. [27] explored the Embodied carbon emission flows in regional industrial chains by combining input–output methods and DEA models to identify the key industrial sectors and key pathways of the Embodied carbon emission flows between regions. This study is a critical breakthrough in regional inter-industry Embodied carbon emission flows, while overlooking the relationship of Embodied carbon emission flows between industrial sectors from an industrial system perspective. A more effective research method that can reveal the dynamic interactions between sectors holistically should be selected to analyze the Embodied carbon flows between industrial sectors from an industrial system perspective. The complex network model is a preferable solution to these problems. It is an effective research method for analyzing the internal structural characteristics of complex systems and can be utilized to uncover the relationships between the internal parts of the system and determine the functions, roles and values of the network nodes. Complex network models are currently adopted in a wide range of fields. Ma et al. [28] investigated some deep features of the network of ore-energy flows in global trade based on complex network theory. Jiang et al. [29] explored the network topology of Embodied energy flows between global industry sectors and derived important evolutionary patterns. Jiang et al. [30] combined a complex network model with the EEIO model to characterize the macroscopic nature of the implicit carbon transfer network in China’s industrial sectors.

As suggested in the above research on Embodied carbon flows, much of the current research focuses on the relationship between Embodied carbon flows in the trade process, involving international trade and domestic trade. Research on Embodied carbon emissions flows from the industrial sector stresses the relationship between industrial sectors at the national level. There is little research on the relationship between industrial sectors and carbon emissions among provinces and regions. On the choice of research method, Moran et al. [31] constructed a multi-regional input–output model covering 41 countries and 35 industries using MRIO and WIOD to calculate the Embodied carbon flows in China’s foreign trade from 1995 to 2011. Lin et al. [10] examined the factors influencing the dynamics of Embodied carbon emissions from the industrial sector in China by combining the I–O model with the LMDI analytical model. These studies demonstrate direct links between carbon emissions from different industrial sectors, but do not effectively capture the indirect links between the various industrial sectors in the industrial system. Moreover, they fail to recognize the position, role and value of the specific industrial sector within the industrial system. The paper aims to answer the following key questions.

(1) How can key carbon-emitting industrial sectors be identified at the provincial level?

(2) How can key Embodied carbon emission flow paths be captured?

(3) How can effective inter-provincial carbon emission reduction, industrial structure optimization and clean production capacity be provided?

With the purpose of filling these gaps, the input–output model and social network analysis method are combined in this research to explore the important position of the industrial sector in China’s industrial development in the three provinces of Heilongjiang, Jilin and Liaoning. Meanwhile, the Embodied carbon emission flow relationships among industrial sectors in the three provinces of Hei-Ji-Liao are analyzed deeply at the provincial level, key carbon-emitting industrial sectors are identified and key Embodied carbon emission flow paths are derived. This paper provides theoretical support and policy formulation reference for reducing carbon emissions, contributing to optimizing industrial structure and improving clean production capacity between provinces.

The remainder of the paper is organized as follows. In Section 2, the model and the data sources are presented. In Section 3, the empirical results are analyzed. In Section 4, the results are discussed and policy implications are provided. Finally, conclusions are drawn in Section 5.

## 2. Research Methods and Data

### 2.1. Research Framework and Model Selection

In this research, an input–output model and a complex network model were chosen to analyze the relationship between embodied carbon emissions flows among industrial sectors in Heilongjiang, Jilin and Liaoning provinces of China, so as to identify the key carbon-emitting industries in each province. Figure 2 illustrates the research framework and model selection for this paper, with specific model construction details described in Section 2.1 and Section 2.2, respectively. Section 2.4 provides the data sources and calculation process.

### 2.2. Data Sources

In this paper, four main types of data sets are used, consisting of input–output (IO) tables, energy balance sheets, energy statistics yearbooks and Guidelines for National Greenhouse Gas Inventories (IPCC). IO tables are published by the statistical bureaus of Liaoning, Jilin and Heilongjiang provinces. The energy balance sheet and the Energy Statistics Yearbook (for Liaoning, Jilin and Heilongjiang provinces) cover eight major fossil energy sources: coal, coking coal, crude oil, gasoline, paraffin, fuel oil and natural gas [32,33]. Based on the relevant parameters in the IPCC Guidelines for National Greenhouse Gas Inventories, the CO_2_ emission factor $\lambda_{k}$ for energy source k can be calculated as:(1)$$\lambda_{k}=NCV_{k}\times CC_{k}\times \frac{44}{12}$$
where $NCV_{k}$ represents the average low-level heat generation of the energy type k; $CC_{k}$ indicates the carbon emission factor of the energy type k; $\frac{44}{12}$ denotes the ratio of the relative molecular mass of CO_2_ to C. The CO_2_ emission factors for each energy source in each province calculated according to Equation (12) are presented in Appendix A. Energy balance sheets and Energy Statistics Yearbook employ the 2021 data set and IO tables for each province use the 2017 data set. Additionally, data processing is performed to merge the 42 industrial sectors in the IO table into 30, so as to ensure that the industrial sector classification in the IO table is consistent with the industrial energy consumption classification in the Energy Statistics Yearbook. The merged industrial sector classification is detailed in Table 1.

### 2.3. Analysis of Input–Output Model

#### 2.3.1. Input–Output Analysis

The input–output analysis is a method based on the Leontief inverse matrix. It is intermediate use plus end use equals total output [34,35], expressed in mathematical notation as:(2)$$\overset{n}{\underset{j=1}{\sum }}x_{ij}+Y_{i}=X_{i}\;\left(i,\;j=1,\;2,\;\cdots ,\;n\right)$$
where $x_{ij}$ represents the value of the production of industry sector i to demand for the product of industry sector j and the value of intermediate inputs in industry sector i is obtained by summing up with j; $Y_{i}$ denotes the amount of end-use value produced by industry sector i to satisfy final needs; $X_{i}$ signifies the total output of industry sector i.

#### 2.3.2. Direct Consumption Coefficient

The industry sector is represented by A as a matrix of direct consumption factors. In matrix A, the element in row i and column j is $a_{ij}$, which indicates the amount of value directly consumed by the production of industry sector i on industry sector j. $a_{ij}$ is calculated as:(3)$$a_{ij}=\frac{x_{ij}}{X_{j}}$$

Shifting the term in Equation (3) gives:(4)$$x_{ij}=a_{ij}X_{j}$$

Substituting Equation (4) into Equation (2) yields:(5)$$\overset{n}{\underset{j=1}{\sum }}a_{ij}X_{j}+Y_{i}=X_{i}$$

In Equation (5), the total output of the n industrial sectors can be obtained as a system of n linear equations. The direct consumption factor matrix A, the end-use matrix Y, the unit matrix I and the total output matrix X are expressed as:$$A=\left[\begin{array}{ccccc} a_{11} & a_{12} & \cdots & a_{1,\;n-1} & a_{1n}\\ a_{21} & a_{22} & \cdots & a_{2,n-1} & a_{2n}\\ \cdots & \cdots & \cdots & \cdots & \cdots \\ a_{n-1,1} & a_{n-1,2} & \cdots & a_{n-1,n-1} & a_{n-1,n}\\ a_{n1} & a_{n2} & \cdots & a_{n,n-1} & a_{nn} \end{array}\right]\;Y=\left[\begin{array}{c} \begin{array}{c} Y_{1}\\ Y_{2}\\ \cdots \end{array}\\ Y_{n-1}\\ Y_{n} \end{array}\right]$$
$$I=[\begin{array}{ccccc} 1 & 0 & \cdots & 0 & 0\\ 0 & 1 & \cdots & 0 & 0\\ \cdots & \cdots & \cdots & \cdots & \cdots \\ 0 & 0 & \cdots & 1 & 0\\ 0 & 0 & \cdots & 0 & 1 \end{array}]\;X=\left[\begin{array}{c} \begin{array}{c} X_{1}\\ X_{2}\\ \cdots \end{array}\\ X_{n-1}\\ X_{n} \end{array}\right]$$

Then Equation (5) can be written as:$$[\begin{array}{ccccc} a_{11} & a_{12} & \cdots & a_{1,\;n-1} & a_{1n}\\ a_{21} & a_{22} & \cdots & a_{2,n-1} & a_{2n}\\ \cdots & \cdots & \cdots & \cdots & \cdots \\ a_{n-1,1} & a_{n-1,2} & \cdots & a_{n-1,n-1} & a_{n-1,n}\\ a_{n1} & a_{n2} & \cdots & a_{n,n-1} & a_{nn} \end{array}]\times \left[\begin{array}{c} X_{1}\\ X_{2}\\ \cdots \\ X_{n-1}\\ X_{n} \end{array}\right]+\left[\begin{array}{c} Y_{1}\\ Y_{2}\\ \cdots \\ Y_{n-1}\\ Y_{n} \end{array}\right]=\left[\begin{array}{c} X_{1}\\ X_{2}\\ \cdots \\ X_{n-1}\\ X_{n} \end{array}\right]$$

Then,
(6)AX+Y=X

Transforming Equation (6) by shifting the terms gives:(7)$$X={\left(I-A\right)}^{-1}Y$$
where ${\left(I-A\right)}^{-1}$ indicates the Leontief inverse matrix, which creates a bridge between end-use and total input. Specifically, either end-use and total input is a constant and the other can be derived.

#### 2.3.3. Competitive Embodied Carbon Emissions Model

CO_2_ emissions from energy consumption in the industrial sector [13,36] are expressed as:(8)$$C_{i}=\overset{n}{\underset{k=1}{\sum }}C_{ik}=\overset{n}{\underset{k=1}{\sum }}\left(\theta_{ik}\times \emptyset_{k}\right)\;\;\;\;\;\;\;\;\;\;\;\;\;\;\;\left(i,k=1,2,\cdots ,n\right)$$
where $C_{i}$ denotes the emissions of CO_2_ from the sum of energy consumption in industry sector i; $C_{ik}$ represents the CO_2_ emissions resulting from the consumption of energy type k by industry sector i; $\theta_{ik}$ signifies the physical amount of energy of type k consumed by industry sector i; $\emptyset_{k}$ stands for the CO_2_ emission coefficient for energy of type k.

The CO_2_ emissions from the direct input energy requirement per unit of output in industry sector i are defined as the direct emission factor [37,38], denoted as $E_{i}\left(i,j=1,2,\cdots ,n\right)$, which is calculated by:(9)Ei=CiXi=∑k=1nCik=∑k=1n(θik×∅k)Xi

The formula for calculating the implied carbon emissions C due to end use is Equation (10):(10)$$C=EX=E{\left(I-A\right)}^{-1}Y\;\;\;\;\;\;\left(E\;is\;a\;determinant\;consisting\;of\;E_{i}.\right)$$

### 2.4. Analysis of ECETN Model Structure

In this section, the ECETN model is constructed and metrics from complex networks are utilized for overall network analysis.

#### 2.4.1. Construction of the ECETN Model

Nodes, edges and weights are key factors in the construction of complex networks [39]. Through the calculations in Section 2.1 and Section 2.2, a complex network for industry sectors is constructed as nodes, the relationship of embodied carbon emission flows between two sectors is constructed as directed edges and the weights of the edges are constructed as embodied carbon emissions, which is ECETN. Furthermore, the structural features of the ECETN model are captured by analyzing the complex network metrics of the ECETN model. In the ECETN model, the number of nodes and edges is based on economic relationships. There is a flow of products or services between nodes and this generates a flow of embodied carbon emissions, as exhibited in Figure 3.

A threshold that can effectively evaluate the relationship of Embodied carbon emission flows is set in this paper to further explore the relationship of Embodied carbon emission flows between industry sectors. When the flow is less than or equal to the threshold, the Embodied carbon emissions flow between industry sectors is negligibly small and then the edge between nodes does not exist [40]. When the flow is greater than the threshold, the Embodied carbon emissions flow between industry sectors is large and is written as an edge:(11)$$f_{ij}\{\begin{array}{c} 1\;\left(e_{ij}>\varepsilon_{i}\right)\\ 0\;\;\left(e_{ij}\leq \varepsilon_{i}\right) \end{array}\;\;\;\;\;\;\;\varepsilon_{i}=\frac{1}{n}\overset{n}{\underset{j=1}{\sum }}e_{ij}$$
where $f_{ij}$ represents the Embodied carbon emission flow relationship between industry sectors; $e_{ij}$ indicates the implied carbon flows between industry sectors i and j; $\varepsilon_{i}$ denotes the threshold value, which is the average Embodied carbon emissions between the two nodes.

#### 2.4.2. Analytical Indicators for the ECETN Model

In this section, the ECETN model is analyzed using complex network metrics, as detailed below.

(1) Centrality Analysis

Centrality, as one of the main focuses of research in complex network analysis, describes the central position that nodes have in the network. In this paper, degree centrality, intermediate centrality and betweenness centrality are used for analysis [6].

① Degree Centrality

Degree centrality refers to the number of other industry nodes directly connected to the industry node. The more other nodes are connected, the more important and influential the node is in the ECETN. The formula is:(12)CRD′(x)=Odx+Idx(2n−2)
where n denotes the network size, $Od_{x}$ indicates the point-out degree of x, $Id_{x}$ refers to the point-in degree of x and $C_{RD}^{\prime }$ signifies the degree centrality of x.

② Betweenness Centrality

According to the American sociologist Freeman (1979), if a node is between several nodes in a complex network, it must have a low degree and this relatively low degree point may play a vital intermediary role. Thus, it is at the center of the network. In ECETN, betweenness centrality is adopted to describe the degree of control that industry nodes have over the flow of Embodied carbon emissions in a complex network. The formula is:(13)$$\;b_{jk}\left(i\right)=\frac{g_{jk}\left(i\right)}{g_{jk}}$$
(14)$$C_{ABi}=\overset{n}{\underset{j}{\sum }}\overset{n}{\underset{k}{\sum }}b_{jk}\left(i\right),\;j\neq k\neq i,\;\;\;j<k$$
where $g_{jk}$ indicates the number of edges between point j and point k; $g_{jk}\left(i\right)$ denotes the number of edges passing through the third point i.

③ Closeness centrality

Closeness centrality describes the distance between an industry node and other nodes in the ECETN. The shorter the distance, the easier it is for the node to Embodied carbon emission flows in the ECETN. Hence, the more likely it is to be at the center of the ECETN, the higher the centrality. It is calculated as:(15)$$C_{APi}^{-1}=\overset{n}{\underset{j=1}{\sum }}d_{ij}$$
(16)$$C_{RPi}^{-1}=\frac{C_{APi}^{-1}}{n-1}$$
where $d_{ij}$ denotes the distance between point i and point j connecting the edges, as well as the number of edges.

(2) Influence analysis

Influence describes the ability of the Embodied carbon emission flows of the nodes in the ECETN to act on other industries. In this paper, the direct consumption factor matrix is employed to assign weights to the adjacency matrix. The KATZ index is utilized to measure the influence of industries, calculated as:(17)$$T=aC+a^{2}C^{2}+\cdots +a^{k}C^{k}+\cdots =\overset{\infty }{\underset{k=1}{\sum }}a^{k}C^{k}={\left(I-aC\right)}^{-1}-I$$
where a represents the decay factor, $C^{k}$ denotes the number of indirect paths of i, j through k (after weighting by the direct consumption factor matrix), $t_{ij}$ refers to the elements of the matrix T, $t_{j}=\sum_{i}t_{ij}$ signifies the column sums and $t_{i}=\sum_{j}t_{ij}$ indicates the row sums.

(3) Clustering Analysis

In ECETN, K-cores, Main Core Networks and clustering coefficients are chosen to complete the screening of key carbon emission industry nodes for the aggregation analysis of Embodied carbon emission flows [29,41,42].

① K-core diagrams are mainly used to reflect the importance and agglomeration of nodes. They are one of the methods to analyze cohesive subgroups of complex networks based on the degree of nodes. K-cores are defined as follows. Let the graph G=(V, E), V be the set of points in graph G and E be the set of edges in graph G (if a is G directed graph, E denotes the set of arcs), defined as W⊆V. Then, the subgraph $H_{k}=\left(W,E|W\right)$ in graph G is a K-cores and also needs to satisfy when and only when ∨v∈W, the degree degree(v)≥k of the point v and $H_{k}$ is the largest subgraph with this feature.

② Main Core Network

Generally, the K-cores subgraph with the largest kernel value is called the Main Core Network ($M_{n}$). In the Main Core Network ($M_{n}$), the industry nodes exhibit strong connectivity and high aggregation. In this paper, the K-cores of the ECETN are first determined. Afterward, the Main Core Network is identified by the maximum kernel value. Finally, the industry nodes in the Main Core Network are the most qualified as key carbon-emitting industries [43].

③ Clustering coefficients

Apart from these two indicators, the Clustering Coefficient is employed to quantitatively describe the degree of aggregation of ECETN. Specifically, a larger Clustering Coefficient indicates a higher degree of aggregation of ECETN [44]. The higher the Clustering Coefficient, the higher the degree of aggregation of the ECETN. The specific calculation formula is:(18)Ci=Eiki(ki−1)2
(19)$$C=\frac{1}{N}\overset{N}{\underset{i=1}{\sum }}C_{i}$$
where $E_{i}$ denotes the actual number of edges that exist between the $k_{i}$ neighboring nodes of node i, $C_{i}$ represents the clustering coefficient of an industry node i of degree $k_{i}$ in the network and C refers to the clustering coefficient of the ECETN.

(4) Diffusivity analysis

In ECETN, key carbon-emitting industries also need to measure the strength of their diffusion ability in the network, that is, the diffusivity of the network nodes, in addition to having a high degree of aggregation. In this research, the strength of association in key carbon-emitting industries is evaluated using the two main indicators of the extent of association and diffusion effects.

① Extent of association

The extent of association is made up of the forward and backward association extents, describing the number of nodes directly connected to other nodes, namely, the number of industries in the ECETN directly supplied/demanded by the Embodied carbon emission flows. In this research, the node’s out-degree ($Od_{i}$) and in-degree ($Id_{i}$) are calculated to quantify the number of other industries in the network directly driven by the industry’s Embodied carbon emissions, so as to measure the importance of the node in the ECETN.

② Diffusion effects

The diffusion effect is a combination of nodes in the Main Core Network examining the primary key carbon-emitting industries, with the alternative nodes pulling or driving other nodes in the periphery through forward and backward linkages. The main two indicators are the forward and backward diffusion effects. The specific calculation formulas are:(20)$$DE_{i\in M}^{\begin{array}{c} output\\ i \end{array}}=\frac{d_{i}}{Od_{i}}=\frac{1}{Od_{i}}\underset{j\in V-M}{\sum }OD_{ij}$$
(21)$$DE_{j\in M}^{\begin{array}{c} iutput\\ j \end{array}}=\frac{d_{j}}{Id_{j}}=\frac{1}{Id_{j}}\underset{i\in V-M}{\sum }ID_{ij}$$
where $DE_{i}^{output}$ indicates the forward diffusion effect of the alternative nodes in the Main Core Network; $DE_{j}^{input}$ denotes the backward diffusion effect of the alternative nodes in the Main Core Network; V−M represents the non-critical node at the periphery of the Main Core Network.

## 3. Empirical Results

### 3.1. Calculation of Embodied Carbon Emission Flows between Industrial Sectors

#### 3.1.1. Direct Carbon Emissions from the Industrial Sector

In this research, the direct carbon emissions and direct carbon emission intensity due to energy consumption by industry sectors in Heilongjiang Province, Jilin Province and Liaoning Province for 2017 are estimated. As suggested in Figure 4a, direct carbon emissions from fossil energy consumption by sector in the three provinces are lowest in Jilin Province and highest in Heilongjiang Province. Electricity & heat production and supply (C24) is the largest direct carbon-emitting sector in Liaoning and Heilongjiang provinces. Petroleum processing and coking (C11) is the largest direct carbon-emitting sector in Jilin province, accounting for 41.2% (Liaoning), 43.6% (Heilongjiang) and 52.7% (Jilin) of total direct carbon emissions in each province. This is followed by metal smelting and rolling processing (C14), which account for 22.2% (Liaoning), 10.9% (Jilin) and 14.6% (Heilongjiang) of total direct carbon emissions by province [45].

In Figure 4b, the industrial sectors with high direct carbon emission intensity in the three provinces are petroleum processing and coking (C11), metal smelting and rolling processing (C14) and production and supply of heat and electricity (C24). These industrial sectors are all basic energy-based sectors. Since the main economic activity in the three provinces of Hei-Ji-Liao is industry, industrial development brings a large consumption of energy such as coal. However, the key industrial sectors that contribute to carbon emissions in each province should be captured, energy use should be effectively restructured and new energy industries should be vigorously developed, so as to lessen the increase in carbon emissions from energy consumption in the three provinces. As observed in Figure 4b, the direct carbon emission intensity of the industrial sectors in the three provinces is stable, except for those with high carbon emissions in C11, C14 and C24. This indicates that the sectors with high carbon intensity are initially controlled to assure the reduction of total carbon emissions in each province.

The direct carbon emissions caused by the consumption of energy in the production process of a particular industrial sector also have an impact on the carbon emissions of upstream and downstream industrial sectors along the chain and intermediate products, forming the flow of Embodied carbon emissions. Total carbon emissions from the industrial sector are composed of direct carbon emissions and Embodied carbon emissions. Considering the dynamics and complexity of the Embodied carbon emission flow process between industry sectors, the research perspective of complex networks is taken.

#### 3.1.2. Embodied Carbon Emissions from the Industrial Sector

The Embodied carbon emissions and the Embodied carbon intensity of the industrial sector in Hei-Ji-Liao provinces were derived from the input–output model. As revealed in Figure 5, Embodied carbon emissions and Embodied carbon intensity fluctuate considerably in three provinces, suggesting that Embodied carbon emission flows between industrial sectors exert a significant impact on their carbon emissions. Particularly, Heilongjiang Province presents the most pronounced performance. In Figure 5a, the Embodied carbon emissions of all eight industrial sectors in Heilongjiang Province exceed their corresponding direct carbon emissions. Figure 5b also reflects the sharpest fluctuations in the fold of its Embodied carbon emissions intensity. From this perspective, the Embodied carbon emission flows between industry sectors should be emphasized to more effectively control the total amount of carbon emissions.

### 3.2. ECETN Model Construction and Topology Analysis

A network diagram and a ring diagram of the Embodied carbon flow network between industry sectors in Hei-Ji-Liao provinces are drawn using Gephi-0.9.2 and Circos-v19 software (Figure 6, Figure 7 and Figure 8) to fully demonstrate the topological structure of the Embodied carbon flow network between industry sectors. Theoretically, there should be 900 input–output relationships between the 30 industrial sectors. Nevertheless, the actual number of relationships existing in the 2017 input–output tables of the three provinces of Hei-Ji-Liao is less than 900, at 605 (Liaoning), 615 (Jilin) and 470 (Heilongjiang). Therefore, the Embodied carbon emission flows between the industrial sectors in each province are measured using the input–output relationships of the three provinces in a ring diagram (Figure 6b, Figure 7b and Figure 8b). Additionally, Embodied carbon flows between industrial sectors are analyzed, reflecting that the production and consumption of goods and services in industrial sectors are separated. The sector’s input and output to itself does not fall into this category. As a result, the sides of the Embodied carbon flow network are 575 (Liaoning), 585 (Jilin) and 440 (Heilongjiang). In Figure 6a, Figure 7a and Figure 8a, the circles represent the industry sectors and the numbers denote the industry sector numbers. The size of the circles represents the point intensity. The larger the circle, the higher the point intensity. The thickness of the edge demonstrates the weight of the implied carbon emission flow. The thicker the edge, the larger the implied carbon emission flow. The arrows signify the flow direction. The structural characteristic indicators of the corresponding ECETNs in Hei-Ji-Liao provinces are calculated using complex network theory. The results obtained are listed in Table 2.

The small-world character of complex networks is measured using clustering coefficient and average shortest path. Specifically, the value of the characteristic path length between points in the network is small, close to a random network, while the aggregation coefficient of the network is high, close to a regular network. The clustering coefficients of the three provincial ECETNs are 0.873 (Liaoning), 0.902 (Jilin) and 0.84 (Heilongjiang), as calculated by Equations (17) and (18). This indicates that the effective association number of the network is high and most industrial sectors are highly clustered. The average shortest paths for ECETN in the three provinces are 1.26 (Liaoning), 1.052 (Jilin) and 1.151 (Heilongjiang), implying that the network has better connectivity, with each industrial sector requiring only 1.26 steps/1.052 steps/1.151 steps to other sectors. Consequently, the three ECETNs have large clustering factors and small average shortest paths, all of which are small-world networks. The small-world character of the ECETN suggests that carbon emission flows between the various industrial sectors in the network are relatively tightly related. The changes in one sector of the network can quickly spill over to other sectors, leading to changes in the entire industrial system. From another perspective, the ECETN is extremely sensitive and vulnerable in certain circumstances, bringing crucial opportunities to collaborative emission reductions between industry sectors in the three provinces of Hei-Ji-Liao.

### 3.3. Analysis of Key Carbon-Emitting Industry Sectors in ECETN

As unveiled by measuring the correlation structure and correlation intensity of the nodes in the ECETN, the key carbon emission industry sector should have the following two characteristics. First, concerning correlation structure, the key carbon emission industry occupies a vital position in the ECETN and can influence the carbon emissions of the industrial system, especially for the carbon emissions of upstream and downstream industries. Second, regarding the intensity of the correlation, the key carbon-emitting industry has a strong ability to gather resources, guaranteeing that a small change in it can lead to changes in the surrounding industries or even the whole ECETN [46]. Additionally, key carbon-emitting sectors have a strong diffusion effect, revealing a diffusion effect to other industries. This assures that emission reduction measures in key sectors can spread to the ECETN and lead to emission reductions in all sectors. In summary, industries that meet the above criteria and have strong potential for development are considered key carbon-emitting industry sectors. In this section, the key carbon-emitting industry sectors in the ECETN are identified through analysis of the mediating role, impact intensity analysis and structural analysis of cohesive subgroups [25,47].

#### 3.3.1. Analysis of the Mediating Role of the Industrial Sector

The degree centrality, intermediate centrality and near centrality are measured based on the concept of node centrality of ECETNs in Hei-Ji-Liao provinces to effectively evaluate the value of industrial nodes in ECETNs, with threshold values of each centrality as Liaoning province (17.073, 1.606, 27.891), Jilin province (14.634, 2.207, 25.786) and Heilongjiang Province (14.634, 1.188, 38.679). This research lists the series of industries with industry sector centrality greater than the threshold value in the ECETNs of the three provinces and considers the industries in the table as strong centrality industry sectors. The details are provided in Table 3, Table 4 and Table 5.

The strong centrality industries in Hei-Ji-Liao provinces in terms of degree centrality, intermediary centrality and proximity centrality are counted.

As suggested in Table 3, the industrial sectors with high Embodied carbon emission spillover capacity (i.e., the capacity of the CO_2_ emitted in the network) in the ECETN of Liaoning province are mostly concentrated in the metal and non-metal products (C15, C13), chemical manufacturing (C12, C25, C19), energy and mining (C2, C3, C4, C5, C24), manufacturing (C6, C17, C19, C20) and public services (C28, C29) sectors. Particularly, the three centrality indicators of C15 and C17 are both high, reflecting the important position of the two industries in ECETN, their strong control over other industries and their Embodied carbon emission spillover ability. They are essential industries worthy of attention.

In the ECETN of Jilin province (Table 4), sectors with high Embodied carbon emission spillover capacity are concentrated in metal and non-metal processing (C13, C14, C15), construction and product manufacturing (C16, C17, C22, C27), chemical manufacturing and others (C12, C23, C26), agriculture, forestry and fishery (C1), energy and mining (C2, C4, C5, C11, C24) and transportation (C18, C28) sectors. Specifically, the two centrality indicators, C16 and C17, are both high, demonstrating the importance of the two sectors in the ECETN and their strong control over other sectors, as well as the carbon emission spillover capacity. They are vital sectors worthy of attention.

In the ECETN of Heilongjiang province (Table 5), sectors with high Embodied carbon emission spillover capacity are concentrated in the agriculture, forestry and fishery (C1), construction (C27), wood products (C9), metal and non-metal products (C13, C14, C15), manufacturing (C16, C22) and energy processing (C12, C24) sectors. Particularly, the two centrality indicators, C1 and C15, are both high, implying the importance of these two sectors in the ECETN and their strong control over other sectors, as well as the ability to spill over carbon emissions. They are critical areas of concern.

The above analysis yields the key industrial sectors in the three provinces of Hei-Ji-Liao with strong Embodied carbon emission spillover capacity and mediating role. Then, the effective control of carbon emissions from these key industrial sectors can influence the flow of Embodied carbon emissions between their upstream and downstream sectors, reduce the flow of implied carbon emissions in the supply chain and thus curtail the carbon emissions of the whole ECETN. Concurrently, the clean development of these key industrial sectors can enhance the efficiency of energy use along the chain, contributing to the lessened carbon emissions of the industrial system.

#### 3.3.2. Impact Intensity Analysis of Industry Sectors

In ECETN, the effect and influence ability of nodes on other nodes is called the point intensity of the industrial sector. In this paper, the 0-1 table is obtained by processing the input–output tables from three provinces of Hei-Ji-Liao. The KATZ index is calculated for the assignment adjacency matrix of the 0-1 table to obtain the point intensity value of each node. As suggested by calculating the impact intensity thresholds of the three provinces as 0.359 (Liaoning), 0.312 (Jilin) and 0.344 (Heilongjiang), the industries greater than this threshold in each province are considered high-impact industries and these industrial sectors carry a considerably large amount of Embodied carbon emissions, as listed in Table 6.

The results of KATZ index statistics in ECETNs (Table 6) demonstrate 23 industrial sectors with KATZ indexes greater than the threshold in Liaoning Province, except for oil and gas extraction, science and technology and services and public services. Additionally, the KATZ indexes of all industries are higher than the threshold (0.359), among which the top 15 industries are construction (C27), transportation and traffic (C18, C28), mining and mining products (C2, C4, C14, C15), energy (C11, C12, C24) and manufacturing (C17, C18, C19). The above industrial sectors as high-impact industrial sectors of implied carbon emissions directly influence the total amount of Embodied carbon emission flows in ECETN. Notably, the impact intensity of agriculture, forestry, animal husbandry & fishery and metal products is much higher than that of other sectors, indicating that these two sectors have a more significant pulling effect on the generation of carbon emissions in other sectors and act as aggregation centers of Embodied carbon emission flows in the network.

In Jilin province, there are 21 industries above the threshold, among which the public service industry (C29, C30) has a weak influence and the top 15 industries are: agriculture, forestry, animal husbandry & fishery (C1), processing and manufacturing (C16), energy and chemical industry (C2, C12, C24, C25, C23, C26), metallurgical manufacturing (C4, C5, C13, C14, C15) and Machinery manufacturing (C16). Among them, the mining industry is at the center of the Embodied carbon emission flow. In other words, coal and non-metal mining are both the supply and consumption centers of Embodied carbon emissions, are in the midstream of the industrial system and serve as crucial producers and suppliers of intermediate products.

In Heilongjiang Province, there are 18 industries above the threshold and the high-impact industries are more concentrated in agriculture (C1), construction (C27), metal products (C13, C15), energy and chemical industries (C3, C10, C12, C14) and manufacturing (C16, C17, C22). Agriculture, construction and manufacturing have always been vital pillar industries in Heilongjiang province, revealing that they exert a strong influence on the industrial system. They carry a large amount of Embodied carbon emissions while impacting the carbon generation and emissions of other industrial sectors.

After the screening results of centrality and influence in ECETNs of Hei-Ji-Liao provinces are compared, industries with strong centrality and high influence were selected as the primary key carbon emission industries. The results are presented in Table 7.

The distribution of the primary key carbon emission industries in the three provinces of Hei-Ji-Liao suggests that agriculture, coal mining and metal products are the most critical Embodied carbon emission industries in the industrial system, coinciding with the industrial structure of northeast China. Since these three industrial sectors are also critical supports for Northeast China’s industries, effective control of carbon emissions from these three industries can facilitate the achievement of carbon emission reduction targets in the regional economic systems of the three provinces.

#### 3.3.3. Correlation and Cohesion Analysis of ECETNs

① K-core and Main Core Network

The direct consumption coefficient matrix of Hei-Ji-Liao provinces was transformed into an adjacency matrix. Then, the K-cores of the adjacency matrix were visualized by Netdraw to obtain K-cores plots, as rendered in Figure 9.

$M_{n}$ is the submap of the ECETN with the largest K-cores, as well as the area with the highest density of correlations between the Embodied carbon emissions flows of the industrial sectors, which are strongly clustered. It is also the area where small changes can significantly lead to changes in the Embodied carbon emissions of neighboring industrial sectors or even industrial systems. Accordingly, the provincial main core networks were extracted and screened for high aggregation industrial sectors with Embodied carbon emission flows. As observed in Figure 9, the maximum core value is 9 for Liaoning Province and 7 for both Jilin and Heilongjiang Provinces, uncovering that the aggregation of Embodied carbon emission flows between industrial sectors in the main core network is slightly stronger in Liaoning than in Jilin and Heilongjiang.

There are 20 industrial sectors in Liaoning’s main core network, mostly concentrated in agriculture and animal husbandry (C1), mineral extraction (C2, C3, C5,), processing and machinery manufacturing (C7, C10, C13, C14, C15, C16, C17, C18, C19, C20), energy and chemicals (C11, C12, C25), distribution services (C28) and public services (C29, C30); 21 industrial sectors in the Jilin main core network, mostly concentrated in agriculture and animal husbandry (C1), mineral extraction (C2, C3, C4, C5), processing and manufacturing and energy (C6, C7, 13, C14, C15, C16, C17, C18, C19), construction and distribution services (C25, C28) and public services (C29, C30); 18 industrial sectors in Heilongjiang’s main core network, mostly concentrated in agriculture and animal husbandry (C1), mineral extraction (C2, C3, C4), processing and machinery manufacturing (C7, C13, C14, C15, C16, C17, C18, C20, C25), energy and chemicals (C12, C23, C25,), distribution services (C28) and public services (C29, C30).

② Clustering coefficients

In addition to analyzing the industry sectors with strong aggregation of implied carbon emissions by determining the main core network using K-cores, clustering coefficients are adopted to determine the degree of aggregation of the industry sectors in the ECETN, with larger clustering coefficients indicating a higher degree of aggregation in the ECETN. The average clustering coefficients of ECETNs in the three provinces were measured to be 0.727 (Liaoning), 0.697 (Jilin) and 0.639 (Heilongjiang). The clustering coefficient thresholds for industry sectors in the three provinces were 0.614, 0.621 and 0.465 for Liaoning, Jilin and Heilongjiang, respectively. These values were for strongly clustered industry sectors. The industrial sectors whose clustering coefficient values were all greater than the threshold in the main core network were those with high aggregation of Embodied carbon emission flows.

The industry sectors with clustering coefficients greater than the threshold were selected as alternative key carbon-emitting industry sectors according to the comparison between the primary key carbon-emitting industry sectors identified above and the highly aggregated industry sectors in the main core network. The final alternative key carbon emission industry sectors for Liaoning are coal mining (C2), chemicals (C12), processing and manufacturing (C15, C17, C19) and distribution services (C28, C29). The alternative key carbon sectors in Jilin are mining and processing (C2, C5, C13, C14), processing and manufacturing (C15, C16), energy and chemicals (C12, C24) and distribution services (C28). The alternative key carbon sectors in Heilongjiang are agriculture and animal husbandry (C1) and metallurgical manufacturing (C13, C14, C15, C16). These are listed in Table 8.

③ ECETNs’ cohesive structure analysis

Cohesive subgroup partitioning of ECETNs is performed based on modular algorithms using Gephi software. The maximum modularity of Liaoning, Jilin and Heilongjiang is 0.426, 0.301 and 0.214, respectively. This result is ideal for the division of cohesive subgroups, as illuminated in Figure 10. The size of the circles in the diagram represents the size of the point intensity of the industrial sector. The larger the circle, the greater the point intensity. The thickness of the edges represents the weight of the Embodied carbon flow. The thicker the edge, the greater the Embodied carbon flow. Figure 10 demonstrates that the nodes in the Liaoning ECETN are divided into two cohesive subgroups, three in Jilin and four in Heilongjiang.

The composition of industrial sectors and Embodied carbon flows within each cohesive subgroup of the three provinces are illustrated in Table 9. According to the analysis of Embodied carbon emission flows within the cohesive subgroups, Liaoning Province has the largest number of flows within subgroup I. Subgroup II has a larger number of industrial sectors, including manufacturing, construction and services and other related sectors; while these sectors are more interrelated and have more complex flow relationships, the overall mobility is not the largest. Although there are not many industrial sectors in Subgroup I, agriculture and coal mining are the pillar industries of the province and are closely related to the upstream and downstream industrial sectors, driving the overall carbon emission transfer volume. There are three cohesive subgroups in Jilin Province, in descending order: III, II and I. Subgroup III contains the largest number of alternative key carbon-emitting industries and their high-centeredness and strong influence result in large inter-sectoral carbon transfers within subgroup III. Meanwhile, subgroup I comprises no alternative key carbon-emitting industries and has the weakest carbon transfer capacity. The Embodied carbon emission flows of the cohesive subgroups in Heilongjiang Province are II, III, I and IV in descending order. The agriculture and livestock sector (C1) in the II cohesion subgroup has a reasonably strong capacity to move carbon emissions. It is a linked industry in several industrial sectors and, as the province’s mainstay industry, should receive focused attention.

Regarding the flow of Embodied carbon emissions between cohesive subgroups, the inflows and outflows of the subgroups in Liaoning province are balanced. Nevertheless, the inflows of subgroup II in Jilin province and subgroup II in Heilongjiang province are much larger than the outflows and are the absorption sites of Embodied carbon emissions in ECETNs. In contrast, the outflows of subgroup I in Jilin province and subgroup III in Heilongjiang province are much larger than the inflows. They have a stronger spillover effect of Embodied carbon emissions and are the dispersal sites of Embodied carbon emissions in ECETNs. The industrial sectors in some cohesive subgroups are in the middle and lower reaches of the chain and their development and growth require large amounts of energy and raw materials from the upstream industrial sectors, resulting in large carbon inflows. The upstream sectors of the chain are the suppliers of raw materials and essential support for the development of the downstream industries, contributing to significant carbon outflows.

#### 3.3.4. Diffusivity Tests for Alternative Key Carbon-Emitting Industry Sectors

In this section, the diffusion of alternative key carbon-emitting industry sectors in ECETNs is examined by province, involving two main components: the breadth of association and the diffusion effect. Particularly, the breadth of association is calculated through the outward and inward degrees of the industry sector. The diffusion effect suggests that the forward diffusion effect, forward diffusion effect, backward diffusion effect and backward diffusion effect of the industry sectors are measured to determine the diffusion of Embodied carbon emissions in ECETNs for alternative key carbon-emitting industry sectors, as illustrated in Figure 11.

Overall, the diffusion effect in Liaoning (0.34) is better than in Heilongjiang (0.15) and Jilin (0.12). The analysis of alternative key carbon-emitting industry sectors demonstrates that, although some alternative key carbon-emitting industry sectors have strong centrality, high influence and high aggregation, Embodied carbon emission flows are only within the industry chain, compared to their weak contribution to the low carbon emission reduction of industry sectors outside the industry chain, and cannot promote cleaner production in the industry system; so they should not be considered as key carbon-emitting industry sectors and, for example, Liaoning (C17, C29), Jilin (C12, C16, C28) and Heilongjiang (C13) should be removed.

In Figure 11, the key carbon-emitting industrial sectors in Liaoning and Heilongjiang exhibit a better diffusion effect. The diffusion effect ratio of key carbon-emitting industry sectors in Liaoning is larger, implying that these sectors promote the rapid flow of Embodied carbon emissions in the industrial system. On the contrary, Jilin’s key carbon-emitting industrial sector is not satisfactory since it only has a unidirectional diffusion effect. Moreover, the diffusion effect is weak and does not significantly contribute to the Embodied carbon emission flow in the industrial system, revealing the difficulty of emission reduction control through the industrial sector in Jilin.

With respect to industry sectors, the Embodied carbon diffusion effects of C2, C15 and C19 in Liaoning are better than those of C12 and C28. In other words, the mining and manufacturing industries play a dominant role in carbon emissions in the industry system and promoting cleaner development of these industries improves the energy efficiency of the industry chain, leading to curtailed carbon emissions in the industry system as a whole. The values of C14 and C15 association breadth and diffusion effects in Jilin are both large, suggesting that metal manufacturing occupies a crucial position in the ECETN. Both its own Embodied carbon emissions and the Embodied carbon emission flows to the industrial chain and even to the industrial system have a huge impact. The key carbon-emitting industrial sectors in Heilongjiang are concentrated in agriculture and livestock (C1) and manufacturing (C14), both of which have strong diffusion capacity and the highest degree of association. Hence, effective control of carbon emissions from C1 and C14 can significantly lessen the overall emissions of the industrial system.

## 4. Discussion

In recent years, China’s rapid economic development has led to an increase in carbon emissions and thus increased pressure on China to reduce these. However, China, as a responsible power, has been advocating a new, fair and reasonable climate governance system at the international level. While developing its economy, China has fulfilled its commitment to energy conservation and emission reduction, especially after the incorporation of “carbon peaking and carbon neutral” into the overall layout of ecological civilization. Concurrently, China has placed more emphasis on environmental protection, green and low-carbon development and clean production to wrestle with the global climate change situation. In the social and economic system, the production process of industrial sectors is accompanied by a large number of carbon emissions and production and consumption between industrial sectors are closely related. In the production process, the industrial sectors constitute a complex network of carbon emissions between them. Energy saving and emission reduction can be achieved by effectively identifying the key nodes in the network and grasping the carbon emission relationship between industrial sectors. In this paper, ECETNs are established through the input–output model and the SNA method. Specifically, the structural characteristics of ECETNs in Hei-Ji-Liao provinces are demonstrated and the key carbon emission industry sectors in each province’s industrial system are identified through indicators such as centrality, influence, agglomeration and diffusion.

Firstly, the results of the overall network characteristics analysis suggest that the ECETNs in the three provinces have significant small-world characteristics. Thus, a close Embodied carbon emission transmission relationship is observed between most industrial sectors. Moreover, the changes in carbon emissions in any one industrial sector bring about a chain reaction of carbon emissions in upstream and downstream related industrial sectors in the industrial chain, which in turn will impact the changes in total carbon emissions in the whole industrial system.

Secondly, the individual structural characteristics of the network reflect that some industrial sectors are characterized by strong centrality and high influence in the ECETNs. In other words, a few industrial sectors play a crucial role in the network. Among them, agriculture, coal mining and metal products are the most critical carbon-emitting industries in the three provinces, coinciding with the industrial structure of the Northeast. Therefore, differentiated low-carbon measures should be adopted during the process of formulating industrial emissions reduction policies according to the positioning of individual industries in ECETNs.

Finally, an analysis of the clustering of the key industrial sectors of interest unveils clear clustering characteristics in the ECETNs of the three provinces. Coal mining (C2), chemicals (C12), processing & manufacturing (C15, C17, C19) and distribution services (C28, C29) in Liaoning; mining & processing (C2, C5, C13, C14), processing & manufacturing (C15, C16), energy & chemicals (C12, C24) and distribution services (C28) in Jilin; agriculture and livestock (C1) and metallurgical manufacturing (C13, C14, C15, C16) in Heilongjiang all have high agglomeration. The diffusivity tests of the high-concentration sectors demonstrate that mining and manufacturing in Liaoning dominate the carbon emissions of the industrial system and the energy efficiency of the industrial chain can be reinforced by promoting the clean development of these sectors, so as to curtail the carbon emissions of the whole industrial system. Jilin’s metal manufacturing industry occupies a key position in the ECETN. Both its own Embodied carbon emissions and the Embodied carbon emissions flowing into the chain or even the industrial system can exert a significant impact. The main carbon-emitting industrial sectors in Heilongjiang Province are concentrated in agriculture and manufacturing. The total emissions of the industrial system can be significantly reduced by effectively controlling their carbon emissions.

Based on the above study, the Embodied carbon emission flow focus pathways are captured following the maximum marginal power relationships of key carbon-emitting industrial sectors in the ECETNs of the three provinces to better accomplish regional emission reduction control. The specific results are provided in Table 10.

There are three critical paths in Liaoning’s ECETN: two starting with coal mining and metal products, one starting with the chemical industry and ultimately all ending with the construction and service sectors. There are four critical paths in Jilin’s ECETN: two paths starting with the energy sectors end up in transport equipment manufacturing; two paths starting with non-metal beneficiation and metal products end up in services. There are two critical paths in Heilongjiang’s ECETN: both paths, starting with metalworking and general equipment manufacturing, end up in the service sector. The key pathways of Embodied carbon emissions in Hei-Ji-Liao are similar, starting from traditional sectors such as mining & energy and ending in the construction and service sectors. Therefore, the key sectors of Embodied carbon emissions and the flow path of Embodied carbon emissions should be revealed when formulating carbon reduction policies. Besides, the role played by each industry sector along the key path should be demonstrated to discover the optimal carbon reduction measures. This will enable the effect of carbon reduction to move from ‘point’ to ‘line’ and from ‘line’ to ‘surface’ and to form regional linkages. On this basis, a carbon tax policy should also be considered. Many countries already have more mature carbon tax policies. China has been enriching and improving its own carbon tax policy since 2012. The ultimate goal is the sustainable development of a low-carbon economy.

## 5. Conclusions

This study is based on data from the input–output table of Hei-Ji-Liao in 2017, the energy balance table in 2021 and the energy statistics yearbook in 2021. The Embodied carbon emission flow relationships in the Hei-Ji-Liao industrial system were measured using input–output models and SNA. The results demonstrated that the Embodied carbon emissions of the three provinces were a key factor in increasing total carbon emissions, while the amount of Embodied carbon emissions was determined by several key carbon-emitting industry sectors. Through the above analysis, an Embodied carbon emission flow network (ECETN) between industry sectors of Hei-Ji-Liao was constructed. An analysis of the overall characteristics of ECETN revealed its small-world character. The individual characteristics of the ECETN were analyzed using complex network technical indicators (centrality, influence, agglomeration, diffusion) to identify the role and position of key carbon-emitting industrial sectors in the network. Following the Embodied carbon emission flow paths between key carbon-emitting sectors, relevant recommendations were provided for the formulation of carbon emission reduction and industrial restructuring policies.

Meanwhile, there are some limitations to this research. Firstly, the Embodied carbon emission flows between industry sectors in the province were only considered in constructing the ECETN, while the Embodied carbon emission flows between regions were neglected. In future research, the Embodied carbon emission flows between industrial sectors in the region will be further explored to ensure regional economic development while promoting regional low carbon emission reduction. Secondly, a large number of service industry sectors were combined in this study for the purpose of industry sector classification due to the limitations of energy consumption data. This failed to provide an understanding of the specific carbon emissions of the sub-sectors. In future research, the service sector will be refined and its sector-specific carbon emissions will be analyzed. Finally, this paper focuses on the analysis of Embodied carbon emissions and emission reduction control in Northeast China. The findings are not generalizable owing to the differences in geographical location and resource endowment. Therefore, multi-regional comparative studies will be conducted in the future to provide more precise policy recommendations for policymakers.

## Figures and Tables

**Figure 1 ijerph-20-02603-f001:**
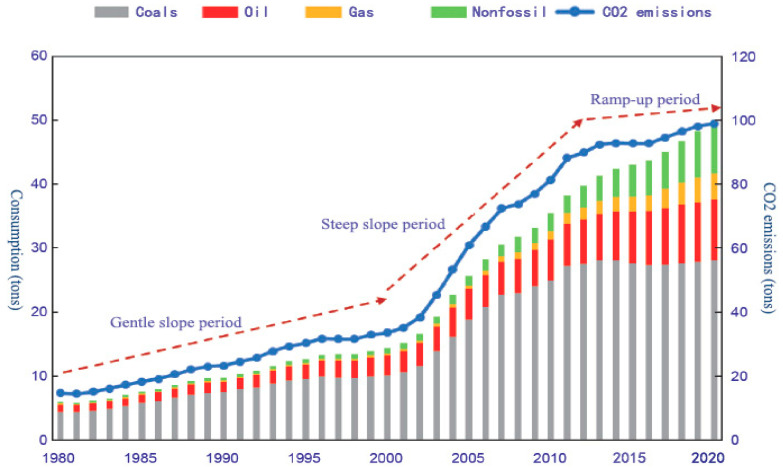
Carbon emission trends.

**Figure 2 ijerph-20-02603-f002:**
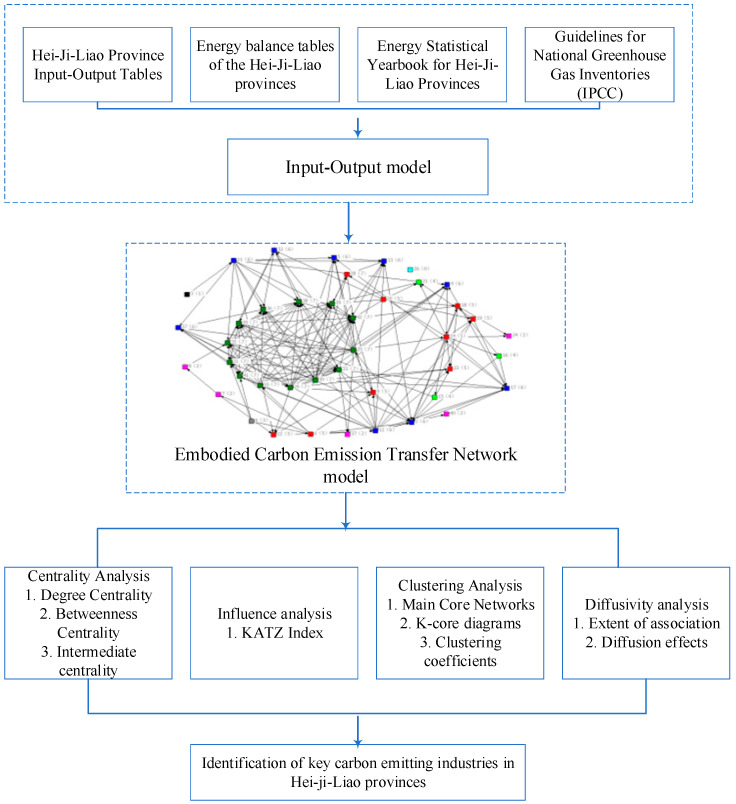
Technical framework and model selection for research.

**Figure 3 ijerph-20-02603-f003:**
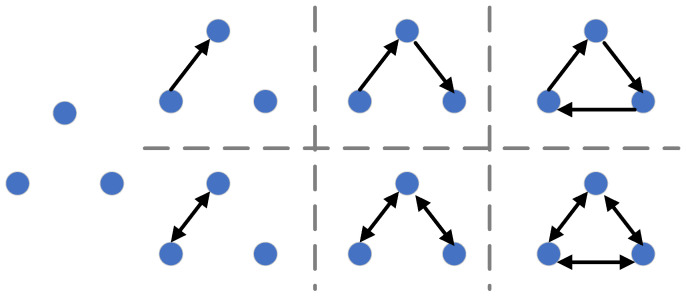
Relationship between nodes and edges in the ECETN model.

**Figure 4 ijerph-20-02603-f004:**
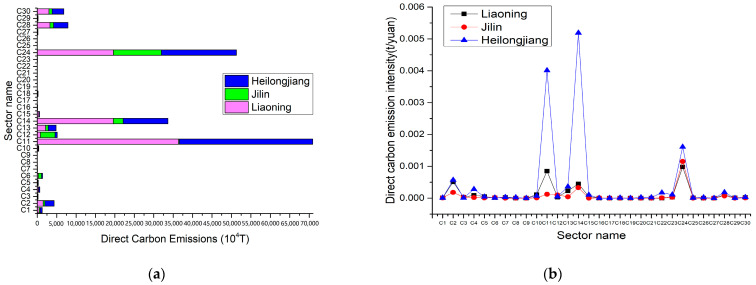
Direct carbon emissions and direct carbon emission intensity due to fossil energy consumption of various industrial sectors in Hei-Ji-Liao. (**a**) Direct carbon emissions of various industrial sectors in Hei-Ji-Liao. (**b**) Direct carbon emission intensity of various industrial sectors in Hei-Ji-Liao.

**Figure 5 ijerph-20-02603-f005:**
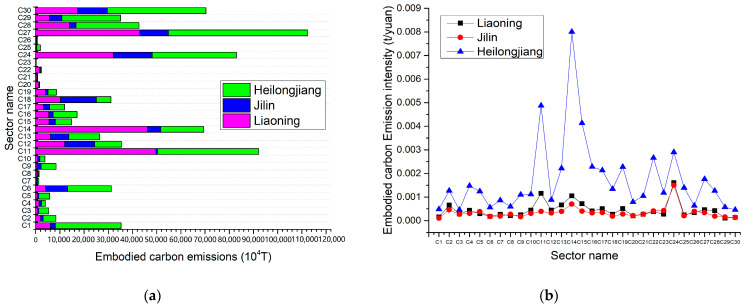
Embodied carbon emissions and Embodied carbon emission intensity due to fossil energy consumption of various industrial sectors in Hei-Ji-Liao. (**a**) Embodied carbon emissions of various industrial sectors in Hei-Ji-Liao. (**b**) Embodied carbon emission intensity of various industrial sectors in Hei-Ji-Liao.

**Figure 6 ijerph-20-02603-f006:**
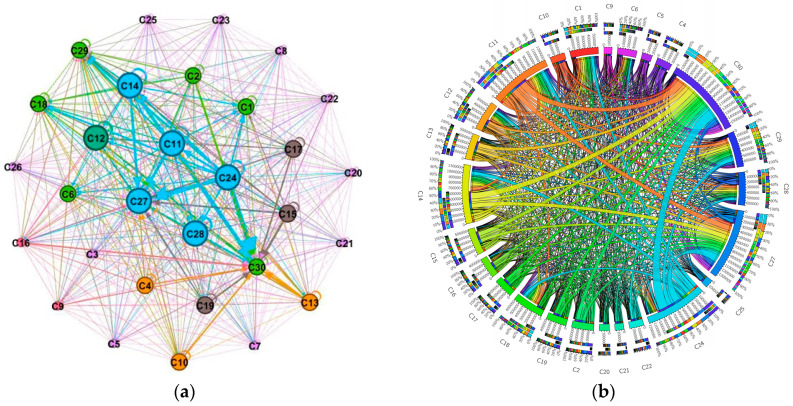
Graph of Embodied carbon emissions flows between industry sectors in Liaoning. (**a**) Embodied carbon emissions network; (**b**) Embodied carbon emissions ring.

**Figure 7 ijerph-20-02603-f007:**
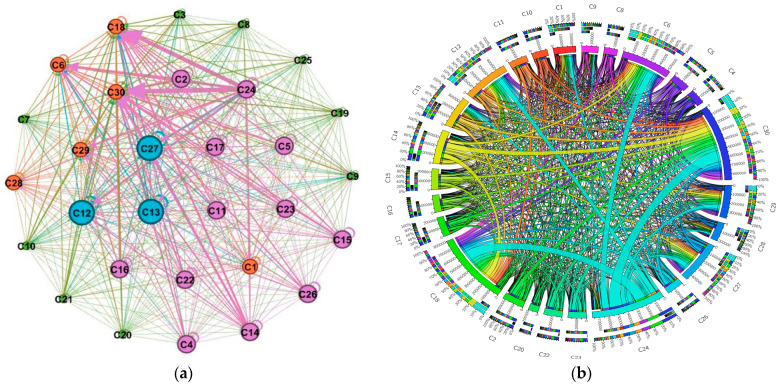
Graph of Embodied carbon emissions flows between industry sectors in Jilin. (**a**) Embodied carbon emissions network; (**b**) Embodied carbon emissions ring.

**Figure 8 ijerph-20-02603-f008:**
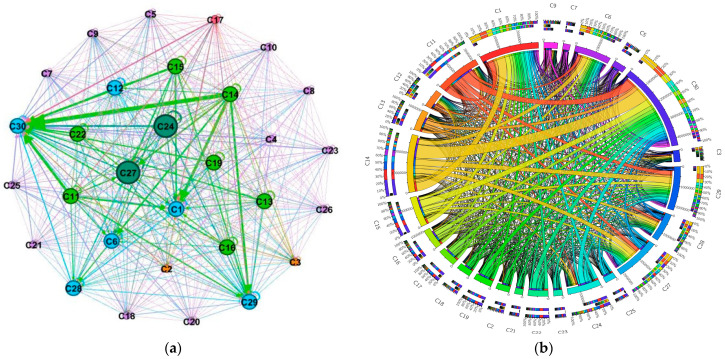
Graph of Embodied carbon emissions flows between industry sectors in Heilongjiang. (**a**) Embodied carbon emissions network; (**b**) Embodied carbon emissions ring.

**Figure 9 ijerph-20-02603-f009:**
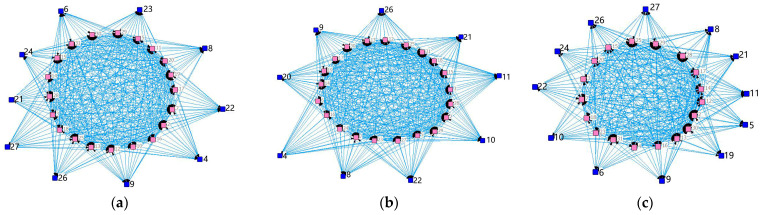
K-cores of the ECETNs in the three provinces of Hei-Ji-Liao. (**a**) K-core for ECETN in Liaoning; (**b**) K-core for ECETN in Jilin; (**c**) K-core for ECETN in Heilongjiang.

**Figure 10 ijerph-20-02603-f010:**
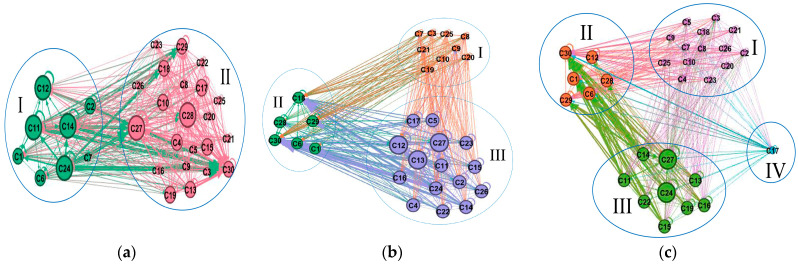
Coalescent subgroups of the ECETNs in Hei-Ji-Liao. (**a**) Coalescent subgroups for ECETN in Liaoning; (**b**) Coalescent subgroups for ECETN in Jilin; (**c**) Coalescent subgroups for ECETN in Heilongjiang.

**Figure 11 ijerph-20-02603-f011:**
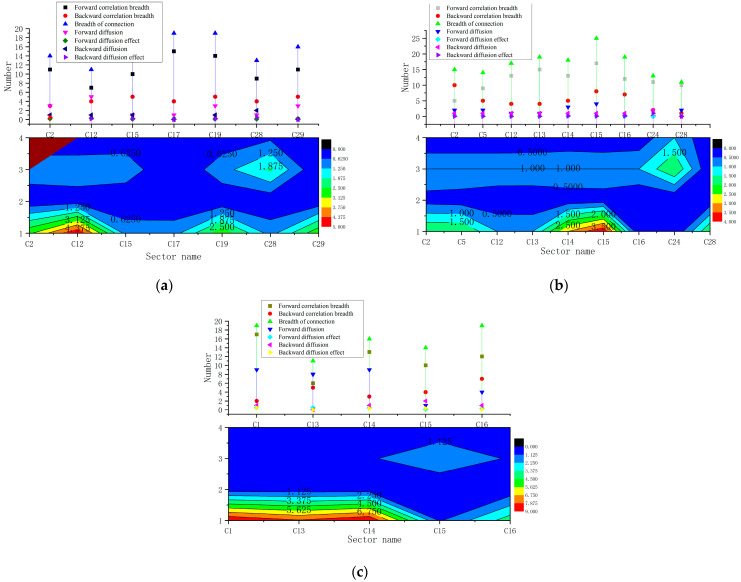
Diffusivity test for alternative key carbon-emitting industry sectors in Hei-Ji-Liao. (**a**) Diffusivity test in Liaoning; (**b**) Diffusivity test in Jilin; (**c**) Diffusivity test in Heilongjiang.

**Table 1 ijerph-20-02603-t001:** Industry Sector Classification.

Code	Industrial Sector	Code	Industrial Sector	Code	Industrial Sector
C1	Agriculture, forestry and fish farming	C11	Petroleum Processing & Coking	C21	Instrument and cultural office machinery manufacturing
C2	Coal Mining & Washing	C12	Chemical industry	C22	Craft & Other Manufacturing
C3	Oil and gas extraction	C13	Non-metallic mineral products	C23	Scrap and waste
C4	Metal mining	C14	Metal Smelting & Rolling	C24	Production and supply of electricity and heat
C5	Non-metallic and other minerals mining	C15	Metalwork industry	C25	Gas production and supply industry
C6	Food manufacture and tobacco processing	C16	General Equipment Manufacture	C26	Water production and supply industry
C7	Textile industry	C17	Specialized equipment manufacturing	C27	Building industry
C8	Textile, clothing, footwear, hats, leather and down and their products	C18	Transport equipment manufacturing	C28	Transport, storage and postal
C9	Wood processing and furniture manufacturing	C19	Electrical machinery and equipment manufacturing	C29	Wholesale, retail, accommodation and catering
C10	Paper, printing and sporting goods manufacturing	C20	Communications equipment and other electronic equipment manufacturing	C30	Other services

**Table 2 ijerph-20-02603-t002:** Results for structural characteristic indicators of ECETNs.

Province	No. of Node	No. of Edge	Average Degree	Network Density	Clustering Coefficient	Average Path Length
Liaoning	30	575	19.17	0.661	0.873	1.36
Jilin	30	585	19.5	0.672	0.902	1.352
Heilongjiang	30	440	16.667	0.506	0.84	1.411

**Table 3 ijerph-20-02603-t003:** Distribution of the values of centrality of 15 industrial sectors in Liaoning Province in 2017.

Industry Sectors	Degree Centrality	Betweenness Centrality	Closeness Centrality
Gas production and supply industry (C25)	36.59	5.75	37.96
Chemical industry (C12)	26.83	1.61	36.28
Metalwork industry (C15)	60.98	16.51	42.27
Non-metallic mineral products (C13)	36.59	1.79	36.94
Wholesale, retail, accommodation and catering (C29)	41.46	2.44	38.68
Transport, storage and postal (C28)	46.34	7.42	39.42
Specialized equipment manufacturing (C17)	73.17	10.25	44.57
Metal mining (C4)	43.90	2.78	38.68
Production and supply of electricity and heat (C24)	31.71	2.51	36.61
Oil and gas extraction (C3)	46.34	3.28	39.05
Food manufacture and tobacco processing (C6)	34.15	1.90	37.62
Electrical machinery and equipment manufacturing (C19)	31.71	1.91	36.94
Coal Mining & Washing (C2)	39.02	2.35	38.32
Communications equipment and other electronic equipment manufacturing (C20)	34.15	2.27	37.27
Non-metallic and other minerals mining (C5)	36.59	2.23	37.62

**Table 4 ijerph-20-02603-t004:** Distribution of the values of centrality of 18 industrial sectors in Jilin Province in 2017.

Industry Sectors	Degree Centrality	Betweenness Centrality	Closeness Centrality
Chemical industry (C12)	17.07	3.27	33.07
Non-metallic mineral products (C13)	31.71	6.43	36.94
Building industry (C27)	26.83	5.93	34.17
Coal Mining & Washing (C2)	39.02	9.18	37.27
Metal mining (C4)	31.71	8.79	35.97
Non-metallic and other minerals mining (C5)	24.39	3.65	35.35
Petroleum Processing & Coking (C11)	17.07	2.24	32.28
Metal Smelting & Rolling (C14)	14.63	3.21	34.17
Metalwork industry (C15)	39.02	8.11	37.62
General Equipment Manufacture (C16)	41.46	4.89	37.27
Specialized equipment manufacturing (C17)	46.34	7.30	39.42
Craft & Other Manufacturing (C22)	31.71	3.46	35.97
Scrap and waste (C23)	34.15	4.67	36.61
Production and supply of electricity and heat (C24)	31.71	2.32	35.65
Water production and supply industry (C26)	31.71	5.27	35.65
Agriculture, forestry and fish farming (C1)	36.59	2.21	37.27
Transport equipment manufacturing (C18)	29.27	3.46	35.35
Transport, storage and postal (C28)	34.15	5.23	36.94

**Table 5 ijerph-20-02603-t005:** Distribution of the values of centrality of 10 industrial sectors in Heilongjiang Province in 2017.

Industry Sectors	Degree Centrality	Betweenness Centrality	Closeness Centrality
Production and supply of electricity and heat (C24)	26.83	1.69	56.94
Building industry (C27)	41.46	2.96	63.08
Petroleum Processing & Coking (C11)	51.22	6.03	67.21
Non-metallic mineral products (C13)	31.71	1.77	58.57
Metal Smelting & Rolling (C14)	39.02	2.51	61.19
Metalwork industry (C15)	95.12	35.42	95.35
General Equipment Manufacture (C16)	58.54	7.27	70.69
Wood processing and furniture manufacturing (C9)	31.71	1.86	59.42
Craft & Other Manufacturing (C22)	48.78	4.66	66.13
Agriculture, forestry and fish farming (C1)	39.02	7.07	62.12

**Table 6 ijerph-20-02603-t006:** Top 15 industry sectors in KATZ index distribution.

Liaoning Province		Jilin Province		Heilongjiang Province	
Code	Value	Code	Value	Code	Value
C27	0.84	C12	1.03	C24	1.15
C28	0.91	C13	0.64	C27	0.61
C12	0.65	C27	0.66	C14	0.65
C24	0.76	C24	0.72	C1	0.71
C14	0.66	C14	0.56	C15	1.78
C11	1.49	C5	1.21	C28	1.24
C15	1.61	C2	1.65	C29	1.84
C4	0.86	C25	0.56	C13	0.66
C17	0.65	C4	0.60	C16	0.71
C19	0.83	C28	0.81	C12	0.69
C30	1.07	C23	0.56	C17	0.64
C1	1.36	C15	0.97	C3	0.94
C29	0.65	C26	1.19	C22	1.47
C18	0.80	C1	0.70	C19	1.11
C2	0.73	C16	0.58	C10	0.87

Note: Calculated from the data of the three-province assignment adjacency matrix.

**Table 7 ijerph-20-02603-t007:** Primary selection of key carbon-emitting industry sectors in Hei-Ji-Liao.

Province	Strong Center Degree Industries	High Impact Industries	Priming Key Carbon-Emitting Industries
Liaoning	(C2, C3, C4, C5, C6, C12, C13, C15, C17, C19, C20, C24, C25, C28, C29), 15	(C1, C2, C4, C11, C12, C14, C15, C17, C18, C19, C24, C27, C28, C29, C30), 15	(C2, C4, C12, C15, C17, C19, C24, C28, C29), 9
Jilin	(C1, C2, C4, C5, C11, C12, C13, C14, C15, C16, C17, C18, C22, C23, C24, C26, C27, C28), 18	(C1, C2, C4, C5, C12, C13, C14, C15, C16, C23, C24, C25, C26, C27, C28), 15	(C1, C2, C4, C5, C12, C13, C14, C15, C16, C24, C26, C28), 12
Heilongjiang	(C1, C9, C11, C13, C14, C15, C16, C22, C24, C27), 10	(C1, C3, C10, C12, C13, C14, C15, C16, C17, C19, C22, C24, C27, C28, C29), 15	(C1, C13, C14, C15, C16, C22, C24, C27), 9

**Table 8 ijerph-20-02603-t008:** Alternative key carbon-emitting industry sectors’ clustering coefficients in Hei-Ji-Liao provinces.

**Province**	**Liaoning**								
**Code**	C2	C12		C15	C17	C19	C28		C29
**Clustering coefficient**	0.64	0.62		0.64	0.85	0.64	0.62		0.64
**Province**	**Jilin**								
**Code**	C2	C5	C12	C13	C14	C15	C16	C24	C28
**Clustering coefficient**	0.64	0.64	0.63	0.62	0.64	0.65	064	0.63	0.65
**Province**	**Heilongjiang**								
**Code**	C1		C13		C14		C15		C16
**Clustering coefficient**	0.77		0.48		0.48		0.48		0.48

**Table 9 ijerph-20-02603-t009:** Description of the structure of the cohesive subgroup in Hei-Ji-Liao provinces.

Cohesive Subgroup No.	Industrial Sector No.	No. of Industrial Sectors	Internal Embodied Carbon Flow (Million Tons)	Embodied Carbon Emissions Outflow (Million Tons)	Embodied Carbon Emissions Inflow (Million Tons)
**Liaoning**					
**I**	C1, C2, C6, C7, C11, C12, C14, C24	8	152,223.32	322,586.00	250,336.40
**II**	C3, C4, C5, C8, C9, C10, C13, C15, C16, C17, C18, C19, C20, C21, C22, C23, C25, C26, C27, C28, C29, C30	22	122,943.29	467,400.10	539,210.30
**Jilin**					
**I**	C3, C7, C8, C9, C10, C19, C20, C21, C25	9	19,791.01	98,883.70	26,735.80
**II**	C1, C6, C18, C28, C29, C30	6	38,268.03	45,572.60	244,693.00
**III**	C2, C4, C5, C11, C12, C13, C14, C15, C16, C17, C22, C23, C24, C26, C27	15	58,576.96	319,424.80	169,809.60
**Heilongjiang**					
**I**	C2, C3, C4, C5, C7, C8, C9, C10, C18, C20, C21, C23, C25, C26	14	32,802.30	468,315.70	145,062.50
**II**	C1, C6, C12, C28, C29, C30	6	147,400.10	136,213.40	1,150,714.80
**III**	C11, C13, C14, C15, C16, C19, C22, C24	9	127,540.40	986,130.80	122,174.40
**IV**	C17	1	6140.20	75,557.10	14,362.20

Note: red markers are alternative key carbon-emitting industrial sectors.

**Table 10 ijerph-20-02603-t010:** Key Embodied carbon emission pathways in Hei-Ji-Liao.

Province	Critical Paths	Flow Volume (10^4^ t)	Proportion
Liaoning	C2-C14-C11-C13-C30	297,181	3.2
	C12-C27-C19-C18-C28-C30	249,304	2.68
	C15-C24-C11-C27	380,153	4.09
Jilin	C2-C24-C29-C6-C18	64,834	2.76
	C5-C17-C27-C29	26,498	1.13
	C24-C13-C27-C6-C18	74,341	3.17
	C15-C14-C12-C29-C30	42,753	1.82
Heilongjiang	C14-C15-C12-C27-C28-C1-C30	244,155	2.63
	C16-C19-C22-C29-C30	78,644	1,04

## Data Availability

The data used to support the findings of this study are available from the corresponding author upon request.

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
