# Peer review of "Identification of Key Carbon Emission Industries and Emission Reduction Control Based on Complex Network of Embodied Carbon Emission Transfers: The Case of Hei-Ji-Liao, China"

_ijerph, 2023, doi:10.3390/ijerph20032603_

Round 1

Reviewer 1 Report

Major comments:

An important topic and good novel analysis. Please make sure to explain the acronyms used as these are not always known to all readers (e.g. line 144 DEA; line 167 MRO and WIOD). Moreover, there are some places where it would be good if you clarified and be a bit more precise on what you mean by a few concepts – e.g. small-world; scale-free; spillover capacity (carbon leakage?); alternative (?) key carbon emitting sectors; and a few others – as these can be misunderstood by the readership. Line 201 – what do you mean by value? Is this monetary or another value/number/unit? I was wondering if you could write a bit about the different years – why did you not use 2017 data for all analysis – I understand the aim to be as current as possible but what does the 5 yr difference mean (5 yrs is a long time nowadays). Maybe consider using GHG Protocol terms (Scope 1,2,3) e.g. when you talk about direct carbon emission (Scope 1) (line 60)? Would it be possible to evaluate with 1 or 2 real-world examples the results derived – i.e. evaluate the model set-up and results with 1-2 company cases? Might also be an idea for another paper? Table 9 – is there another way to make the inter-comparison more intuitive rather than a table with large numbers - e.g. the overall carbon emission transfer volume (leakage)? In general, the graphics are very hard to see, read and discern. Good suggestions for next studies in the conclusions section.

Minor comments:

Line 44-46: Delete as this is not really science relevant and valued – what is enough and to whom and why?

Figure 1: Units are missing

Line 113: Reference Cole & Elliott (2021) is missing in the list.

Line 174-5: Role and value – not sure what you mean?

Section 2.4: Could you obtain emission factors directly from one or more energy suppliers – maybe for a case study?

Line 399: Instead of only – then say initially (as all sectors eventually will need to de-carbonize).

Line 485: Value (monetary or?)

Line 727-733: Same as the first minor comment – too many value laded inputs (responsible; fair; fulfilled) – it would be sufficient to state that China is working on, like the majority of the rest of the world, transitioning toward a more sustainable and climate friendly economy from line 730 to 733.

Line 795: These analysis could I assume also be used to consider carbon taxation strategies as well as informing emerging Sustainable Finance Taxonomies – not just in China but in other countries? So, that the work and methods presented are considered outside China – how can other countries use this for their purposes on e.g. taxes and taxonomy?

Author Response

Answer to Respected Reviewer Comments

Dear Respected Reviewer,

Hello

Thank you very much for your attention. We also thank you for devoting your valuable time for reviewing our manuscript. The authors are sure that your comments improved the quality of the manuscript. We tried to answer all your comments carefully.

Reviewer#1:

  1. Please make sure to explain the acronyms used as these are not always known to all readers (e.g., line 144 DEA; line 167 MRO and WIOD).

Thanks to the reviewer for your valuable comments.

Following your suggestion, DEA, MRO and WIOD have been added to Nomenclature, which is at the top of the article.

  1. Moreover, there are some places where it would be good if you clarified and be a bit more precise on what you mean by a few concepts – e.g., small-world; scale-free; spillover capacity (carbon leakage?); alternative (?) key carbon emitting sectors; and a few others – as these can be misunderstood by the readership.

Thanks to the reviewer for your valuable comments.

Based on your suggestions, we have specified terms that were unclear in the article, such as Line 451-454 small-world characteristics, Line 21-23 scale-free characteristics, and Line 499-500 spillover capacity, etc.

  1. Line 201 – what do you mean by value? Is this monetary or another value/number/unit? I was wondering if you could write a bit about the different years – why did you not use 2017 data for all analysis – I understand the aim to be as current as possible but what does the 5 yr. difference mean (5 yrs. is a long time nowadays).

Thanks to the reviewer for your valuable comments.

The explanation of this issue is as follows. An input-output table is a balance sheet of linkages between industrial sectors, which reflects the interlinkages and balanced proportional relationships between sectors in a given period. Input-output tables are divided into physical tables and value tables according to different units of measurement. Value in this paper refers to the value of finished goods flowing between industrial sectors and is without a specific unit.

The year of data selection is based on a five-year period of input-output table compilation, i.e., statistical compilation is carried out every five years of compilation, with extensions of the table every two years. This is the official data released by the China Statistics Bureau. Currently, the latest edition of the published input-output table data is the 2017 version, but based on your suggestion, we will keep track of the data and continue to complete the research when the data is updated.

  1. Maybe consider using GHG Protocol terms (Scope 1,2,3) e.g., when you talk about direct carbon emission (Scope 1) (line 60)? Would it be possible to evaluate with 1 or 2 real-world examples the results derived – i.e., evaluate the model set-up and results with 1-2 company cases? Might also be an idea for another paper?  Table 9 – is there another way to make the inter-comparison more intuitive rather than a table with large numbers - e.g., the overall carbon emission transfer volume (leakage)?  In general, the graphics are very hard to see, read and discern.  Good suggestions for next studies in the conclusions section.

Thanks to the reviewer for your valuable comments.

Based on your suggestions for direct carbon emissions in Line60, we have added literature in order to better illustrate this issue. The content of the data inside Table 9 corresponds directly to Figure 10.

  1. Line 44-46: Delete as this is not really science relevant and valued – what is enough and to whom and why?

Thanks to the reviewer for your valuable comments.

We have removed the invalid statements Line44-46.

  1. Figure 1: Units are missing

Thanks to the reviewer for your valuable comments.

We have modified Fig.1 to add units

  1. Line 113: Reference Cole & Elliott (2021) is missing in the list.

Thanks to the reviewer for your valuable comments.

We have added the missing references

  1. Line 174-5: Role and value – not sure what you mean?

Thanks to the reviewer for your valuable comments.

Based on your suggestions, we have highlighted in the text Line 175-177 the reasons for choosing the three provinces of Northeast China, Heilongjiang, Jilin and Liaoning. The three provinces in particular have a rich and dense industrial sector, especially heavy industry and state-owned enterprises, and occupy an important position in China.

  1. Section 2.4: Could you obtain emission factors directly from one or more energy suppliers – maybe for a case study?

Thanks to the reviewer for your valuable comments.

According to your suggestion, we strongly agree with you, but since this paper is mainly a quantitative study and all considered variables are derived from input-output tables, in our future research, we will pay more attention to combining quantitative research with qualitative research and enriching multi-case comparative analysis, which will be the key direction of our research.

  1. Line 399: Instead of only – then say initially (as all sectors eventually will need to de-carbonize).

Thanks to the reviewer for your valuable comments.

We have changed the content of Line399.

  1. what do you mean by punishment and reward?

Thanks to the reviewer for your valuable comments.

We hope that the embodied carbon emissions will attract the attention of the national authorities and strengthen the regulation. This could be rewarded or penalized according to the carbon emissions of the industrial sector, promoting the development of relevant enterprises towards green energy efficiency and clean production.

  1. Line 485: Value (monetary or?).

Thanks to the reviewer for your valuable comments.

Value in this paper refers to the value of finished goods flowing between industrial sectors and is without a specific unit.

  1. Line 727-733: Same as the first minor comment – too many values laded inputs (responsible; fair; fulfilled) – it would be sufficient to state that China is working on, like the majority of the rest of the world, transitioning toward a more sustainable and climate friendly economy from line 730 to 733.

Thanks to the reviewer for your valuable comments.

This part has been amended as you requested.

  1. Line 795: This analysis could I assume also be used to consider carbon taxation strategies as well as informing emerging Sustainable Finance Taxonomies – not just in China but in other countries? So, that the work and methods presented are considered outside China – how can other countries use this for their purposes on e.g., taxes and taxonomy?

Thanks to the reviewer for your valuable comments.

This is a very good comment and we have added this to Line 796-798.

Reviewer 2 Report

The subject matter of the article is interesting and worth describing. The method of execution requires minor adjustments. In the Introduction, the authors presented an introduction to the subject. The Introduction section has some shortcomings. It does not contain all the necessary elements. The main goal is not clearly defined, and there are no specific goals. Research hypotheses or research questions should be given in the Introduction section.

The working layout is correct. Minor adjustments are required. In section 2, the data sources should be presented before the model assumptions. I suggest the following order in this section:

2.1. Research framework and model selection

2.2. data sources.

2.3. Analysis of the Input-Output model.

2.4. Analysis of ECETN model structure.

There is a separate Discussion section, but it needs to be completed. I understand discussion as referring to other research after presenting your research results. In my opinion, conducting research without a clear comparison and reference to other studies makes it impossible to properly assess the results obtained. It is enough to provide a few or a dozen items referring to the research. As it stands, the list of literature is not extensive. It can be expanded a bit.

The Conclusion section is incomplete. One must certainly refer to the hypotheses put forward. Can the hypotheses be tested positively or negatively? Conclusions can be scored. Conclusions should be a synthesis.

Figure 1. What units were used? The axes should have units specified.

Figures 4a and 4b. The font is too small. The axes are illegible. It needs to be improved. The same is true for Figures 5a, 5b, 6b, 7b, 8b, 11a, 11b, 11c.

In general, the article is well prepared, contains a logical sequence and analysis results. A comprehensive analysis has been prepared. After taking into account the comments, the article should be appropriate.

Author Response

Answer to Respected Reviewer Comments

Dear Respected Reviewer,

Hello

Thank you very much for your attention. We also thank you for devoting your valuable time for reviewing our manuscript. The authors are sure that your comments improved the quality of the manuscript. We tried to answer all your comments carefully.

Reviewer#2:

  1. In the Introduction, the authors presented an introduction to the subject. The Introduction section has some shortcomings. It does not contain all the necessary elements.  The main goal is not clearly defined, and there are no specific goals.  Research hypotheses or research questions should be given in the Introduction section.

Thanks to the reviewer for your valuable comments.

Based on your suggestions, we have revised the introduction to clarify the background, significance, objectives and research questions of the study.

  1. In section 2, the data sources should be presented before the model assumptions. I suggest the following order in this section:2.1. Research framework and model selection; 2.2.  data sources.; 2.3.  Analysis of the Input-Output model; 2.4.  Analysis of ECETN model structure.

Thanks to the reviewer for your valuable comments.

Based on your suggestions, we have restructured the subsections in section2.

  1. There is a separate Discussion section, but it needs to be completed. I understand discussion as referring to other research after presenting your research results. In my opinion, conducting research without a clear comparison and reference to other studies makes it impossible to properly assess the results obtained.  It is enough to provide a few or a dozen items referring to the research.  As it stands, the list of literature is not extensive.  It can be expanded a bit.

Thanks to the reviewer for your valuable comments.

Based on your suggestions, we have revised Section 4 and modified it to Discussion.

  1. The Conclusion section is incomplete. One must certainly refer to the hypotheses put forward. Can the hypotheses be tested positively or negatively? Conclusions can be scored.  Conclusions should be a synthesis.

Thanks to the reviewer for your valuable comments.

This study focuses on the use of input-output models and SNA to complete the key industries of embodied carbon emissions in the industrial sectors of Heilongjiang, Jilin and Liaoning in the three northeastern provinces of China and to give optimization paths. No specific assumptions are made as this paper identifies key industries through a complex network indicator review. Based on your suggestions, we have revised the conclusion section to include mainly conclusions, contributions and shortcomings to make it more informative.

  1. Figure 1. What units were used? The axes should have units specified.

Thanks to the reviewer for your valuable comments.

We have modified Fig.1 to add units.

  1. Figures 4a and 4b. The font is too small. The axes are illegible. It needs to be improved. The same is true for Figures 5a, 5b, 6b, 7b, 8b, 11a, 11b, 11c.

Thanks to the reviewer for your valuable comments.

Based on your suggestions, we have adjusted the pixel resolution of the images. This was a key issue as the images would be displayed distorted in the docx format, so the results were very poor. The technical editor was also contacted and in the subsequent typesetting we will provide an eps file suitable for the PDF version, which will be able to guarantee the clarity of the images.
